



# Carbon Cycle Extremes Accelerate Weakening of the Land Carbon Sink in the Late 21st Century

Bharat Sharma[1,2], Jitendra Kumar[3], Auroop R. Ganguly[1], and Forrest M. Hoffman[2,4]

[1]Sustainability and Data Sciences Laboratory, Department of Civil and Environmental Engineering, Northeastern University, Boston, Massachusetts, USA
[2]Computational Sciences & Engineering Division and the Climate Change Science Institute, Oak Ridge National Laboratory, Oak Ridge, Tennessee, USA
[3]Environmental Sciences Division, Oak Ridge National Laboratory, Oak Ridge, Tennessee, USA
[4]Department of Civil and Environmental Engineering, University of Tennessee, Knoxville, Tennessee, USA

**Correspondence:** Bharat Sharma (bharat.sharma.neu@gmail.com)

**Abstract.** Rising atmospheric $CO_2$ concentrations enhance vegetation growth through increased carbon fertilization and water-use efficiency. Terrestrial vegetation takes up more than one-quarter of global anthropogenic carbon emissions, and uptake is modulated by regional climate. Increasing surface temperature could lead to enhanced evaporation, reduced soil moisture availability, and more frequent droughts and heat waves. The spatiotemporal co-occurrence of such effects further drives

extreme anomalies in vegetation productivity and net land carbon storage. However, the impacts of climate change on extremes in net biospheric production (NBP) over longer time periods are unknown. Here we show that due to climate warming, about 88% of global regions will experience a larger magnitude of negative NBP extremes than positive NBP extremes toward the end of 2100, which accelerate the weakening of the land carbon sink. Our analysis indicates the frequency of negative extremes associated with declines in biome productivity was larger than positive extremes, especially in the tropics. While the overall

impact of warming at high latitudes is expected to increase plant productivity and carbon uptake, high-temperature anomalies increasingly induce negative NBP TCEs toward the end of the 21st century. We found soil moisture anomalies as the most dominant individual driver of NBP extremes and the compound effect of hot, dry, and fire caused extremes at more than 50% of the total grid cells. The larger proportion of negative NBP extremes raises a concern about whether the Earth is capable of increasing vegetation production with growing human population and rising demand for plant material for food, fiber, fuel, and

building materials. The increasing proportion of negative NBP extremes highlights the consequences of not only reduction in total carbon uptake capacity but also of conversion of land to a carbon source.

**Short summary**

Rising atmospheric carbon dioxide increases vegetation growth and causes more heatwaves and droughts. The impact of such climate extremes is detrimental to terrestrial carbon uptake capacity. We found that due to overall climate warming, about 88%

of the world's regions towards the end of 2100 will show anomalous losses in net biospheric productivity (NBP) rather than gains. More than 50% of all negative NBP extremes were driven by the compound effect of dry, hot, and fire conditions.





## 1 Introduction

Rising anthropogenic carbon dioxide ($CO_2$) emissions are leading to increases in Earth's surface temperature and climate variability as well as intensification of climate extremes. Terrestrial ecosystems have historically taken up a little over one-quarter

of these emissions via carbon accumulation in forest biomass and soils (Friedlingstein et al., 2019) and helped constrain increasing atmospheric $CO_2$ concentrations. The increase in the net terrestrial carbon sink is a result of reduced deforestation, enhanced vegetation growth driven by $CO_2$ fertilization, and lengthening of growing seasons in high latitudes. The growing terrestrial carbon sink provides a negative feedback to climate change; however, exacerbating environmental changes and climate extremes, such as droughts, heatwaves and fires, have the potential to reduce regional carbon stocks and moderate carbon

uptake (Reichstein et al., 2013; Sharma et al., 2022). Net biospheric production (NBP), the total downward flux of carbon from the atmosphere to the land, represents the net carbon uptake after accounting for carbon losses from plant respiration, heterotrophic respiration, fire, and harvest (Bonan, 2015) and is a critical measure of land carbon storage. Climate-driven large anomalies in NBP could impact the structure, composition, and function of terrestrial ecosystems (Frank et al., 2015). To improve our understanding of the climate–carbon cycle feedbacks, especially during such large carbon anomalies, we investi-

gated the changing magnitude, frequency, and spatial distribution of NBP extremes over decadal time periods and identify the influential climate anomalies that potentially drive large NBP extremes at regional and global scales.

Terrestrial carbon cycle processes, such as photosynthesis, respiration and elemental cycling control the structure, composition and function of terrestrial ecosystems. In the past few decades, the global terrestrial carbon cycle has taken up 25–35% of the $CO_2$ emissions from anthropogenic activities such as fossil fuel consumption, deforestation and other land use changes

(Piao et al., 2019). With rising atmospheric $CO_2$, the carbon uptake by both the land and the ocean has also increased but with significantly greater variability over land (Friedlingstein et al., 2019). The interannual variability in land carbon uptake is strongly influenced by climate extremes, and it is primarily responsible for the interannual variation in the atmospheric $CO_2$ growth rate (Piao et al., 2019).

Climate extremes are part of Earth's climatic variability, affecting terrestrial vegetation and modifying ecosystem-atmosphere

feedbacks (von Buttlar et al., 2018). Recent studies have investigated the influence of rising temperatures on climate extremes and terrestrial ecosystems (von Buttlar et al., 2018; Diffenbaugh et al., 2017; Frank et al., 2015; Zscheischler et al., 2018; Sharma et al., 2022). Observations and climate models suggest that climate change has increased the severity and occurrence of the hottest month, hottest day, and driest and wettest periods (Diffenbaugh et al., 2017). Heavy precipitation or lack thereof could have a negative feedback on the carbon cycle via soil water-logging or drought stress, respectively (Reichstein et al.,

2013). A few studies have investigated the impact of climate extremes on the carbon cycle and found that hot and dry extremes reduce land carbon uptake, especially in low latitudes and arid/semi-arid regions (Pan et al., 2019; Frank et al., 2015). Attribution studies infer that the compound effect of multiple climate drivers has a larger effect on the carbon cycle and its extremes (Sharma et al., 2022; Zscheischler et al., 2018; Pan et al., 2019; Frank et al., 2015; Reichstein et al., 2013) than any individual climate driver. Most attribution methods focus on analyzing the response of the carbon cycle to climate, aggregated over

annual, sub-annual, and seasonal scales; however, the responses may vary over shorter time scales, including daily to monthly.





The variability in climate–carbon cycle feedbacks is dependent on geographical location, among other factors. Grose et al. (2020) reported that while Australia is expected to experience an overall reduction in precipitation by 2100, the spatial distribution of precipitation varies since a few regions are expected to get more and others will receive less precipitation. Ault (2020) found that despite the overall increase in precipitation and water-use efficiency globally, the available soil moisture may be reduced across many regions due to increased evapotranspiration from higher temperatures, exceeding the supply from precipitation. The regions that see a decrease in supply and increase in demand for water are sensitive even to relatively small increases in temperature. These feedbacks will increase the severity of droughts, and ENSOs may further amplify the effect (Ault, 2020). Net primary production (NPP) sensitivity to temperature is negative above 15°C and positive below 10°C (Pan et al., 2019), which means warming will cause a reduction in carbon uptake in the tropics and extra-tropics and an increase in carbon uptake at higher latitudes. However, with increasing average surface temperatures, the NPP sensitivity could become negative over time in high latitude regions.

Rising atmospheric $CO_2$ levels and climate change could have implications for biological (Frank et al., 2015) and ecological systems since the severity and occurrence of climate extremes, such as heatwaves, droughts, and fires, are likely to strengthen in the future. These systems are more sensitive to climate and carbon extremes than to gradual changes in climate. The increasing frequency and magnitude of climate extremes could reduce carbon uptake in tropical vegetation, reduce crop yields (Ribeiro et al., 2020), and negate the expected increase in carbon uptake (Reichstein et al., 2013). In this study we investigated the extremes in NBP and their climate drivers from Earth system model simulations for the period 1850–2100 across several regions around the globe. The objectives of this research were to 1) quantify the magnitude, frequency, and spatial distribution of NBP extremes, 2) attribute individual and compound climate drivers of NBP extremes at multiple time lags, and 3) investigate the changes in climate–carbon cycle feedbacks at regional scales.

## 2 Methods

### 2.1 Data

We used the Community Earth System Model (version 2) (CESM2) simulations at $1° \times 1°$ spatial and monthly temporal resolution to analyze the carbon cycle extreme events in net biospheric production. The CESM2 is a fully coupled global Earth system model composed of atmosphere, ocean, land, sea ice and land ice components. The simulations analyzed here were forced with atmospheric greenhouse gas concentrations, aerosols and land use change for the historical (1850–2014) and Shared Socioeconomic Pathway 8.5 (SSP5-8.5; 2015–2100) scenario, wherein atmospheric $CO_2$ mole fraction rises from about 280 ppm in 1850 to 1150 ppm in 2100 (Danabasoglu et al., 2020). While $CO_2$ forcing causes temperatures to increase, changes in aerosols and land use have a slight cooling effect (Lawrence et al., 2019), resulting in an overall increase of about 8°C in mean air temperature over the global land surface during 1850–2100.



## 2.2 Definition and Calculation of Extreme Events

The Intergovernmental Panel on Climate Change (IPCC) (Seneviratne et al., 2012) defines extremes of a variable as the subset of values in the tails of the probability distribution function (PDF) of anomalies. Based on the global PDF of NBP anomalies, we selected a threshold value of $q = 5$, such that total positive and negative extremes constitute 5% of all NBP anomalies (Figure S1). The negative and positive extremes in NBP were comprised of NBP anomalies smaller than $-q$ and larger than $q$, respectively. While the total number of NBP extremes was constant (i.e., 5% of all NBP anomalies) for any time period, the count and intensity among positive and negative extremes vary, depending on the nature of the PDF of NBP anomalies.

We computed extremes for every 25-year period from 1850 through 2100 to analyze the changing characteristics of NBP extremes at regional to global scales. For regional analysis, we used the 26 regions defined in the IPCC Special Report on Managing the Risks of Extreme Events and Disasters to Advance Climate Change Adaptation (Seneviratne et al., 2012), hereafter referred to as the SREX regions (Figure S2). We analyzed the characteristics of NBP extremes during carbon uptake periods, when photosynthesis dominates NBP, and land is a net sink of carbon (NBP $> 0$), and carbon release periods, when NBP is dominated by respiration and disturbance processes, and land is a net source of carbon (NBP $< 0$) (Marcolla et al., 2020).

The anomalies in NBP were calculated by removing the modulated annual cycle and nonlinear trend from the time series of NBP at every grid cell. We calculated the modulated annual cycle and nonlinear trend of NBP using singular spectrum analysis, which is a non-parametric spectral estimation method that decomposes a time series into independent and interpretable components of predefined periodicities (Golyandina et al., 2001). The conventional way of computing annual cycle or climatology does not capture the intrinsic non-linearity of the climate–carbon feedback (Wu et al., 2008). The modulated annual cycle, composed of signals with return periods of 12 months and its harmonics, is able to capture the varying modulation of the seasonality of NBP under rising atmospheric $CO_2$ concentrations. The nonlinear trend is comprised of return periods of 10 years and longer, such that the anomalies in the ecosystem and climate drivers capture the effects of ENSO, which has a large impact on climate and the carbon cycle (Zscheischler et al., 2014; Ault, 2020). Thus, NBP anomalies consist of intra-annual variability, represented by high-frequency signals (<12 months), and the interannual variability (>12 months and <10 years).

## 2.3 Attribution to Climate Drivers

While most studies traditionally attribute carbon cycle impacts to changes in climate at seasonal to annual time scales, many carbon cycle responses to climate variability occur at shorter, daily to monthly, time scales (Frank et al., 2015). Some recent studies have performed attribution by comparing the median of the climate driver distribution in a large space-time dimension (Zscheischler et al., 2014; Flach et al., 2020), which may not capture the variability at regional to grid cell scales. Using linear regression of time-continuous NBP extremes that represent the large intra-annual and interannual variation in NBP with climate anomalies, we quantified the dominance (regression coefficient) and response (sign of the regression coefficient) of climate drivers on large NBP extremes. The NBP extreme events could either occur contiguously over time and space, or be isolated from other events. Long duration time-continuous extreme events have a larger impact on the terrestrial ecosystem





than time-separated isolated extreme events. As described by Sharma et al. (2022), we define time-continuous extreme (TCE)
events such that they fulfill the following conditions, (i) they must consist of isolated extremes that are continuous for at least
one season (i.e., 3 months) and (ii) any number of isolated or contiguous extremes can be a part of a TCE event if the gap
between such extremes is less than one season in length (i.e., up to 2 months). We assume that extreme events separated by
gaps greater than or equal to one season length are separate TCE events.

Human activities, such as fossil fuel emissions and land use changes, modify biogeochemical and biogeophysical processes,
which alter the climate through climate–carbon feedbacks. Large anomalous changes in climate drivers have a strong impact
on carbon uptake and biospheric productivity. Here, we attributed NBP TCEs to climate drivers, namely, precipitation ("Prcp"),
soil moisture ("SM"), monthly average daily temperature ("TAS"), and biomass loss from carbon flux into the atmosphere due
to fire ("Fire"). As the terrestrial vegetation has ingrained plasticity to buffer and push back effects of climate change (Zhang
et al., 2014), the impacts of changes in climate drivers are often associated with lagged responses. Moreover, the strength of
130 the impact of climate on NBP is dependent on location, timing, and land cover type (Frank et al., 2015). Linear regressions
of TCEs in NBP and anomalies of every climate driver were identified at all affected land grid cells for lags from one to four
months. We assumed that the higher the Pearson Correlation coefficient ($\rho$) of a climate driver with NBP extremes, the larger is
its impact on NBP at that location. The attribution based on $\rho$ is used only for those grid cells for which the significance value
$p < 0.05$. The grid cells with at least two negative and positive NBP TCEs each often yielded high correlation coefficients with
135 high significance values ($p < 0.05$); thus, this constraint was applied for attribution to climate drivers.

The instantaneous impact (when lag equals zero months) of driver anomalies ($dri_t$) on NBP TCEs ($nbp_t$) is computed using
Equation 1, where $N$ represents the number of months in TCEs at any grid cell. Attribution based on the lagged response of
driver anomalies on NBP TCEs was computed using Equation 2, where $L$ represents the number of lagged months. For lags
greater than one month, we computed the correlation of the average of climate driver anomalies, $\frac{dri_{t-l}}{L}$ for every time-step in
the driver anomalies, with $nbp_t$. The resulting $\rho$ captures the average response of antecedent climatic conditions up to $L$ months
in advance that drive NBP TCEs.

for lag = 0:

$$\rho = corr(dri_t, nbp_t) \mid t\epsilon N \tag{1}$$

for lag >0:

$$\rho = corr(\sum_{l=1}^{l=L} \left( \frac{dri_{t-l}}{L} \right), nbp_t) \mid t\epsilon N \tag{2}$$

The direction and strength of the impact of various climate drivers on plant productivity and the carbon sink vary with
space and time. Increased temperatures could lead to increased respiration and losses in NBP in the tropics and mid-latitudes,





but an increase in temperature could conversely lead to higher photosynthetic activity and greater uptake at higher latitudes. A moderate reduction in precipitation may not severely impact vegetation productivity, but if accompanied by a heatwave, it could lead to large losses in NBP. We analyzed the dominant climate drivers across SREX regions for every 25-year period from 1850 to 2100 to understand the changing characteristics of large spatio-temporal extremes and their drivers across time and space.

Anomalous climate drivers causing NBP extremes may or may not qualify as climate extremes by themselves. A recent study found that the periods of extreme climate and NBP often do not occur at the same time, and the compound effect of non-extreme climate drivers could produce an extreme in NPP (Pan et al., 2019). Occurrence of a NBP extreme is also likely driven by the compound effect of multiple climate divers, we identified co-occurring anomalous climatic conditions during and antecedent to NBP extremes to improve our understanding of the interactive compound effect of drivers on the carbon cycle. Since the dominance of climate drivers are usually quantified by a correlation coefficient range of 0.5–0.7 (Dormann et al., 2013), we imposed a limit of correlation coefficients $\rho > 0.6$ and significance values $p < 0.05$ on co-occurring individual climate drivers to qualify as individual or compound drivers of NBP extremes. These constraints yield a smaller number of extremes that are attributable to climate drivers with high confidence.

## 3   Results and Discussion

### 3.1   Characteristics of NBP Extremes

The 5[th] percentile NBP anomalies computed for every 25-year period from 1850 to 2100 rendered threshold trajectories that increase from 140 GgC/month to 220 GgC/month (Figure 1(a)). This 1.5 times increase in threshold values demonstrates the increasing magnitude of anomalies and interannual variability of NBP across the globe. The corresponding time series of intensity of losses and gains in biome productivity were calculated by integrating the negative (NBP anomalies $< -q$) and positive (NBP anomalies $> q$) extreme anomalies. The rate of increase in the magnitude of negative extremes ($-834$ MgC/month) was larger than that of the positive extremes ($804$ MgC/month) (Figure 1(b)), which implies that over time the net losses in carbon storage during NBP extremes increases.

The changes in NBP are driven by spatial and temporal variations in climate drivers and anthropogenic forcing. During 1850–1874, 24 out of 26 regions were dominated by carbon release and the total NBP was negative (Figure S3). From 1850 through the 1960s, the land experienced a net $CO_2$ loss to the atmosphere likely driven by deforestation, fire and land-use change activities (Friedlingstein et al., 2019) (Figure S9). After the year 1960, the continued increase in fossil fuel emissions raised the atmospheric $CO_2$ concentration despite declining rates of deforestation. Increasing $CO_2$ fertilization, water-use efficiency, and the lengthening of growing seasons enhanced vegetation growth and raised NBP with the largest increases in the tropics and northern high latitudes (Figure S4). After 2070, the total NBP reached its peaks and started to decline (Figure S3) as ecosystem respiration exceeded total photosynthetic activity. Tropical regions have the largest magnitude of NBP; however, the rate of increase of NBP declined after 2050 and the region of Sahara (SAH) showed an early drop in total NBP after the year 2050. Longer dry spells and intense rains due to changing precipitation patterns in Mediterranean and subtropical ecosystems

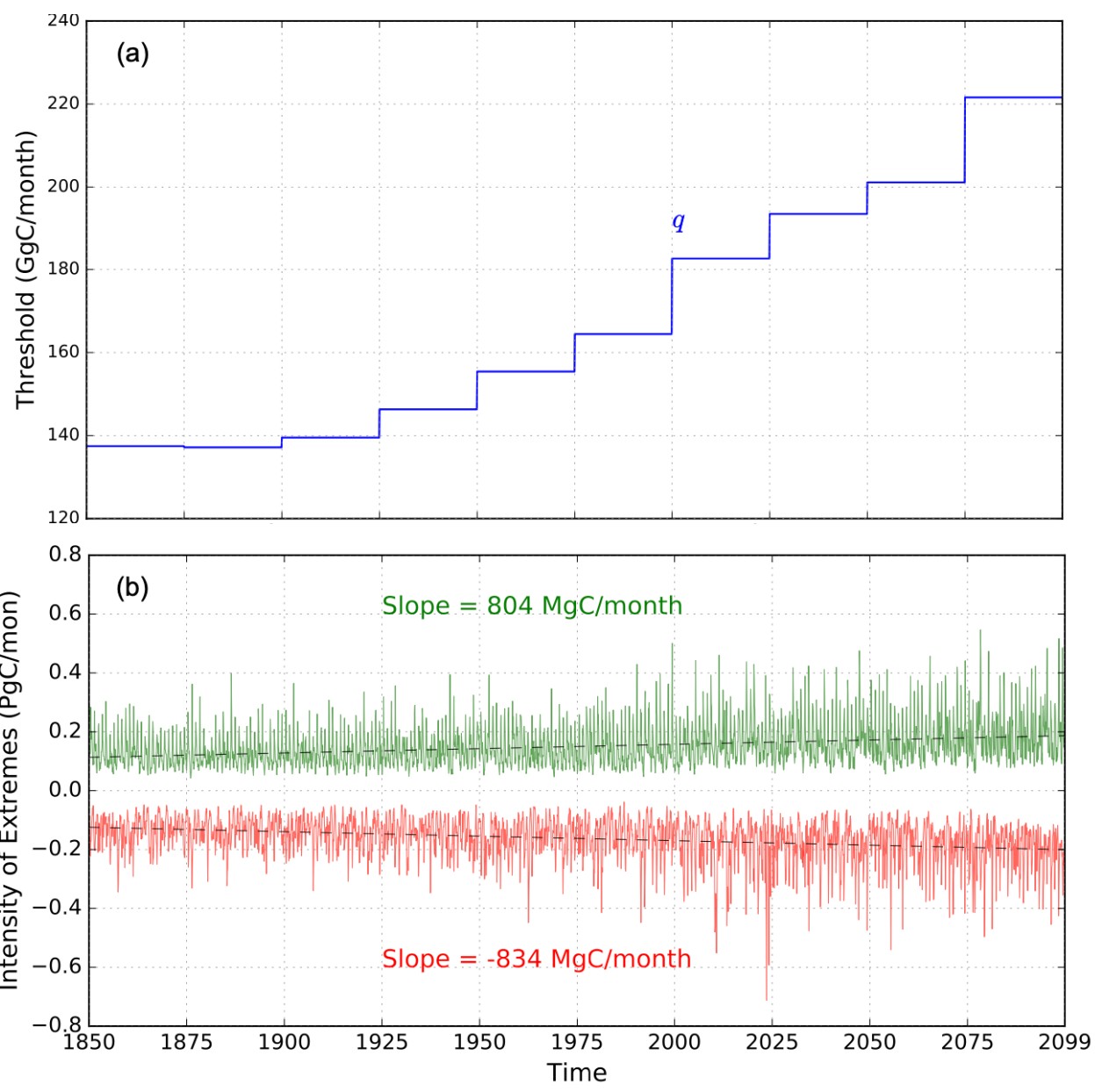

**Figure 1.** (a) The 5$^{th}$ percentile threshold, $q$, of NBP anomalies. The negative extremes in NBP are those NBP anomalies that are $< -q$ and positive extremes are $> q$. (b) The intensity of positive and negative extremes in NBP in CESM2 are represent by green and red color, respectively. The rate of increase of positive and negative extremes in NBP are 804 and $-834$ MgC per month, respectively.





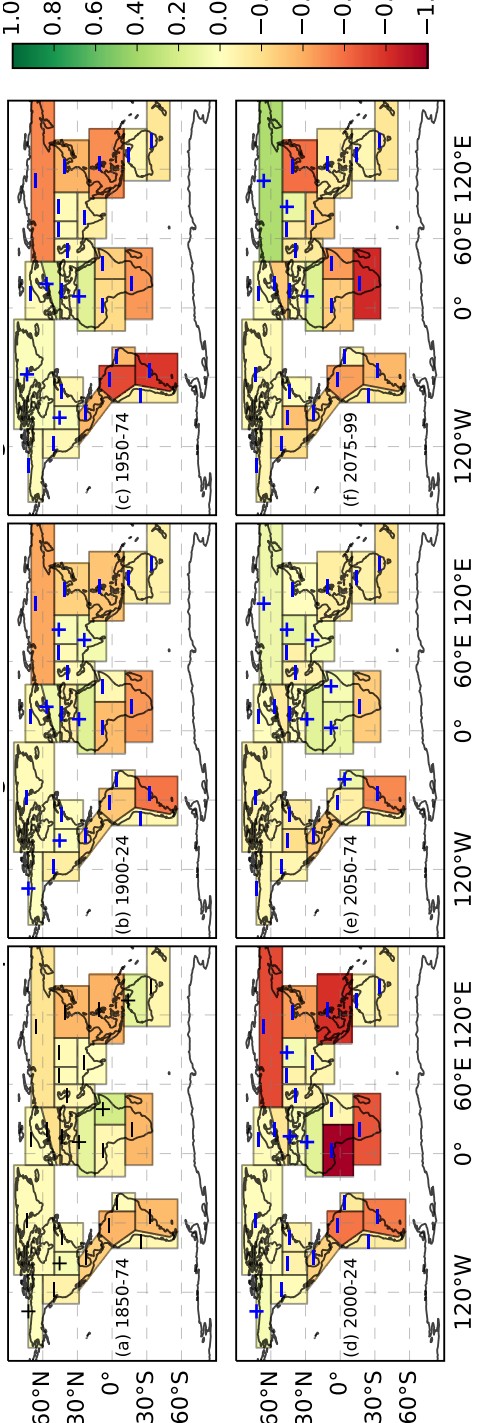

**Figure 2.** The sum of positive and negative carbon cycle extremes is referred as Net Uptake Change during NBP extremes. The figure shows the changing spatial distribution of net uptake change (PgC) during the following periods: (a) 1850–74, (b) 1900–24, (c) 1950–74, (d) 2000–24, (e) 2050–74, and (f) 2075–99. A net gain in carbon uptake is represented by a green color and a '+' sign, and a net decrease is represented by a red color and a '−' sign. For most regions, the magnitude of negative NBP extremes or losses in carbon uptake are higher than positive NBP extremes or gains in carbon uptake.



are likely to cause higher tree mortality (Frank et al., 2015). Hot temperatures and reduced activity of RuBisCO hindering carboxylation are possible factors that will likely cause a net decrease in NBP in the region of SAH and make it a net carbon source after 2050. During 2050–74 and 2075–99, low-latitude regions exhibit the highest regional NBP; however, many areas in the tropics exhibited a declining growth rate of regional NBP (Figure S4).

As anomalous changes in climate vary over space and time, extremes in NBP also respond to the interactive effects of climate

drivers and exhibit spatial and temporal variation. Figure 2 shows the net total sum of both positive and negative extremes in NBP in SREX regions integrated for all 25-year periods (1850–74, 1900–24, 1950–74, 2000–24, 2050–74, and 2075–99). Most regions exhibited net losses in biospheric productivity during extremes, e.g., South Africa (SAF) has always been dominated by negative NBP extremes. The large magnitude of net carbon uptake changes during the period 2000–24 was likely driven by a change in LULCC forcing from decadal to annual during 2000–2015 and then back to decadal for 2015 onward. The change

in resolution of LULCC forcing possibly caused higher climate variability due to biogeophysical feedbacks and subsequently led to increased carbon cycle variability and extremes. 23 out of 26 SREX regions experienced an overall loss in biospheric productivity during extremes near the end of the 21$^{st}$ century (Figure 2). The distribution of the total magnitude and count of negative TCEs during 2075–99 across all the SREX regions followed a similar pattern, i.e., more frequent extremes were accompanied by larger carbon losses (Figure 3(a)). The largest losses in carbon uptake during TCEs were in tropical regions,

e.g., East Asia (EAS), Amazon (AMZ), and SAF, with −3, −3, and −2.25 PgC carbon losses, respectively, during 2075–99. These regions also witnessed the highest number of negative NBP TCEs at 1270, 1410, 950, respectively. The magnitude of carbon losses and the number of negative NBP TCEs were highest in tropical regions. The magnitude and number of negative TCEs were very low for the high latitude regions of Alaska (ALA), Canada and Greenland (CGI), Eastern North America (ENA), Northern Europe (NEU), Central Europe (CEU), and the dry regions of the Mediterranean (MED) and Sahara (SAH).

Although the number of NBP TCEs in NAS were more than in Southeastern South America (SSA) and Central America (CAM), the magnitude of NBP TCEs in NAS were low due to lower regional NBP. Since the extremes were calculated based on global anomalies, the largest impacts on terrestrial carbon uptake are expected in the regions of AMZ, EAS, and SAF, which have the largest concentrations of live biomass.

The magnitude and the total number of regions dominated by negative extremes in NBP are expected to gradually increase in

the 21$^{st}$ century (Figure 3(b)). Most of the increase in the frequency of negative extremes in NBP are expected in ENA, South Asia (SAS), SAF, and CAM (Figure S5). The increase in the magnitude (23 out of 26 or 88% of all regions) and frequency (18 out of 26 or 70% of all regions) of negative NBP TCEs in most SREX regions during 2075–99 is a matter of concern since the total global NBP peaked at around 2070 and subsequently declines in the model (Figure S3). The negative NBP TCEs dominate in eight out of the nine tropical regions, which store the largest standing carbon biomass and represent the largest

portion of carbon uptake loss during negative NBP extremes. A large magnitude of extreme events in the NBP could potentially lead to a state of low and decreasing carbon sink capacity that could further lead to a positive feedback on climate warming. The strengthening of negative extremes relative to positive extremes in NBP represents a net decline of terrestrial carbon sink capacity into the future (Figure S3). The positive feedback of warming and climate driven losses in carbon uptake raise concerns about the implications of reducing terrestrial uptake capacity on food security, global warming, and ecosystem functioning.

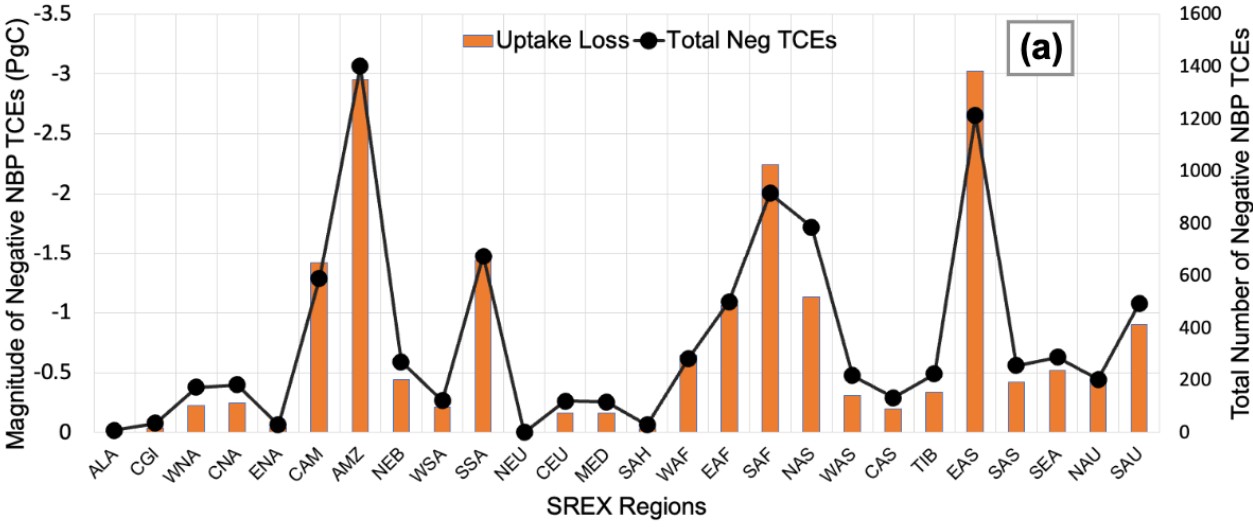

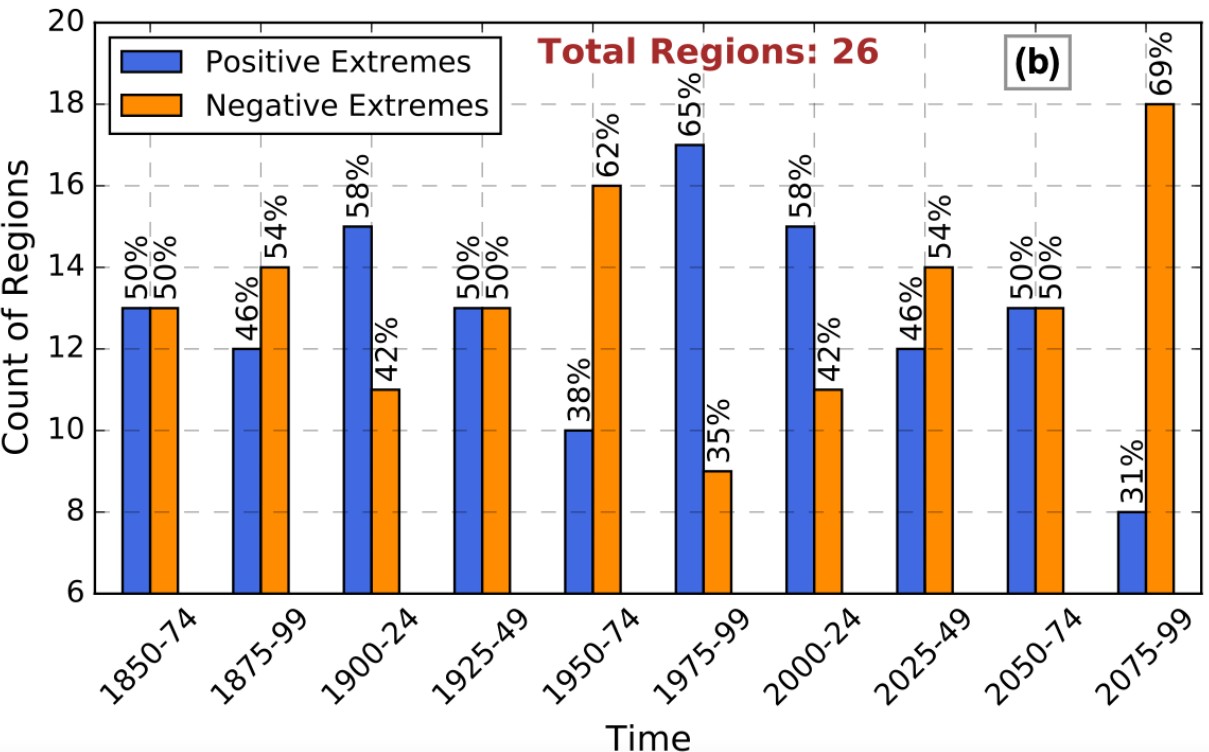

**Figure 3.** (a) Total magnitude of negative carbon cycle extremes or loss is carbon uptake during TCEs across SREX regions plotted as a bar graph (left $y$-axis). The total number of negative TCE events (right $y$-axis) plotted as line graph. The largest portion of carbon uptake loss is in the tropical SREX regions of the Amazon (AMZ), East Asia (EAS), and South Africa (SAF) for the period 2075–99. (b) Count ($y$-axis) of the regions dominated by either positive or negative NBP extremes. Relative to a total of 26 SREX-regions, the percent count of positive or negative NBP extremes is represented at the top of the bars.





Moreover, the sensitivity of vegetation responses is higher for climate and carbon extremes than for gradual changes because of larger response strength and shorter response times (Frank et al., 2015).

## 3.2 Attribution to Climate Drivers

The increase in climate variability and extremes driven by rising $CO_2$ emissions influences the terrestrial carbon cycle (von Buttlar et al., 2018; Reichstein et al., 2013; Sharma et al., 2022). The control of climate drivers on NBP extremes is dependent
on the regional interannual variability of climatic conditions and vegetation composition. The percent of total number of grid cells that show soil moisture as a dominant driver of NBP TCEs were about 40 to 50% from 1850 to 2100 across multiple lags, which means that the near term and long term impact of soil moisture were highest among all other drivers (Figure S6). The positive response of soil moisture anomalies on NBP TCEs indicates that a decline in soil moisture causes a reduction in NBP and vice-versa. Likewise, the dominant climate driver across the 26 SREX regions was also soil moisture and it exhibited a
positive response relationship with NBP TCEs (Figure 4). However, the proportion of the total number of grid cells dominated by precipitation doubled (10 to 20%) when the lag was increased from 1 to 3 months. This implies that antecedent declines in precipitation limit carbon uptake more than a recent decline in precipitation and possibly causes a decline in soil moisture. Moreover, the plants with deep roots are less impacted with short-term reduction in precipitation than prolonged droughts, which are caused by soil moisture limitation. By the end of the 21$^{st}$ century, the model indicates that 70% of the total number
of NBP extremes will be water-driven, i.e., due to soil moisture and precipitation. Our results are consistent with recent studies (Liu et al., 2020; Pan et al., 2019) that concluded that the most important factor limiting vegetation growth is water stress, which is caused mainly by low soil moisture. Lack of soil moisture for extended periods could result in drought events, causing a larger reduction in ecosystem productivity and a smaller reduction in terrestrial respiration.

The second most dominant driver of NBP TCEs was fire, which has a positive response on NBP TCEs (Figure S6). Fire is
an important Earth system process that is dependent on vegetation, climate, and anthropogenic activities. CESM2 incorporated a process-based fire model, which contains three components, namely, fire occurrence, fire spread, and fire impact (Li et al., 2013). The interannual variability of agricultural fires is largely dependent on fuel load and harvesting; deforestation fires included fires due to natural a nd anthropogenic ignitions, caused by deforestation, land-use change, and dry climate. Peat fires usually occur in the late dry season and are strongly controlled by climate. The current version of the fire model reasonably
simulates burned area, fire seasonality, fire interannual variability, and fire emissions (Li et al., 2013). As fires are controlled by soil moisture, temperature, and wind, the attribution of NBP extremes to fires could also include the NBP extremes that are driven by inadequate soil moisture and hot temperatures. Therefore, the total number of fire events could be larger, and recovery after such fire driven extremes could be much longer.

Hot temperatures that persist for long periods induce heatwaves, which tend to reduce ecosystem production and enhanced
terrestrial respiration, causing a large reduction in NBP (Pan et al., 2019). Leaf photosynthesis depends on the RuBisCO-limited rate of carboxylation, which is inversely proportional to the $Q_{10}$ function of temperature in the model (Lawrence et al., 2019). Hubau et al. (2020) found that with increasing temperatures and droughts, tree growth was reduced and could offset earlier productivity gains. Conversely, warm temperatures in the northern high latitudes cause an increase in carbon uptake due



to reduced snow cover and optimal temperature for photosynthesis. Increased warming at northern high latitudes could lead to
hot temperature-related hazards and alter temperature–carbon interactions, which is discussed in Section 3.5.

Rising $CO_2$ emissions drive high temperature in the tropics and have the potential to hinder photosynthesis and vegetation
growth, further discussed in Section 3.4. The changes in atmospheric circulation patterns might also influence the precipitation
cycle, resulting in longer dry spells and increased fire risks with stomatal closure (Frank et al., 2015; Langenbrunner et al.,
2019). The second-largest negative NBP extremes were experienced by arid and semi-arid regions with mostly grasslands
(Figure 3(a)). Several studies conclude that soil moisture causes an increase in dry days and have a significant negative effect
on the carbon cycle driven by increasing droughts in arid, semi-arid, and dry temperate regions (Frank et al., 2015; Zscheischler
et al., 2014; Pan et al., 2019). The regions of South Africa, Central America, and Northern Australia witness the largest NBP
extremes driven by fire. Extremes in the Amazon region were dominated by fire, soil moisture, and temperature in the 21[st]
century. Reduction of fuel load by changing vegetation composition could likely be the reason for fewer fire-dominated regions
later in the 21[st] century.

## 3.3   Compound Effect of Climate Drivers

The interactive effect of multiple climate drivers could lead to devastating ecological consequences as compound extremes
often have a larger impact on the carbon cycle than the aggregate response of individual climate drivers (Zscheischler et al.,
2018; Ribeiro et al., 2020; Pan et al., 2019). We used three broad classes of climate drivers, namely, moisture (dry vs. wet),
temperature (hot vs. cold), and fire to study their compound effect. At most grid cells, NBP extremes were either positively
correlated with anomalous precipitation and/or soil moisture, and/or negatively correlated with temperature and/or fire. Figure 5
shows the compound climate drivers, both mutually exclusive and inclusive, that control NBP extremes over time. Mutually
exclusive drivers are those climatic conditions that do not occur at the same time to cause an extreme event. When the drivers
of an extreme are overlapping they are called mutually inclusive. For example, if an extreme is driven by both *hot* and *dry*
conditions, the mutually exclusive climate driver is only *hot & dry* and the mutually inclusive drivers are *hot*, *dry*, and *hot &
dry*.

The largest fraction, about 50%, of total NBP TCEs were attributed to the combined effect of *hot* & *dry* & *fire* events
(Figure 5). This implies that every other large extreme event associated with anomalous loss in biospheric productivity was
driven by the interactive effect of water limitation, hot days (heat waves), both of which together could trigger fire and rapid loss
of carbon. The second strongest exclusive compound driver was *dry* & *fire*, causing about 25% of extremes. With increasing
climate warming, the number of NBP extremes driven by hot and dry climatic conditions have increased, with about 10%
extremes driven exclusively by *hot* & *dry* events during 2050–74.

Although the negative impact of water limitation (*dry*) on NBP extremes was the highest (driving inclusively about 90% of
all NBP extremes), rising atmospheric $CO_2$ concentration and climate change led to an increasing number, 54% during 1900–
24 to 62% during 2050–74, of NBP extremes driven inclusively by *hot* climatic conditions (Figure 5). For the same periods,
extremes driven inclusively by *hot* & *dry* rose by 8%.



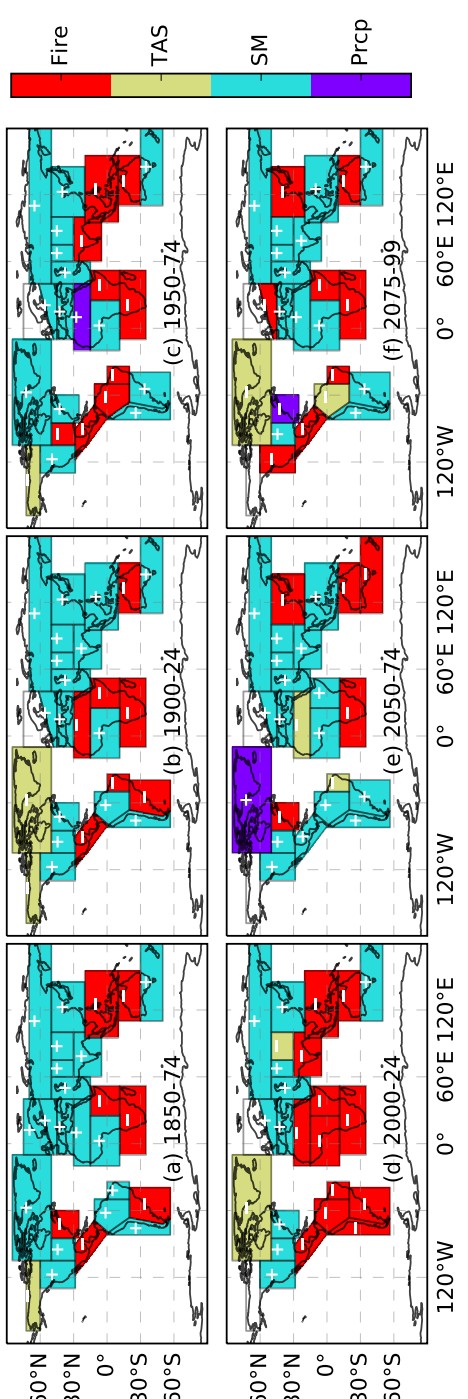

**Figure 4.** Spatial distribution of dominant climate drivers across SREX regions. The color in every SREX region represents the most climate driver causing carbon cycle extremes at 1 month lag for following periods: (a) 1850–74, (b) 1900–24, (c) 1950–74, (d) 2000–24, (e) 2050–74, and (f) 2075–99. The positive ('+') and negative ('−') sign within a region represents the correlation relationship of NBP extremes with every dominant climate drivers.





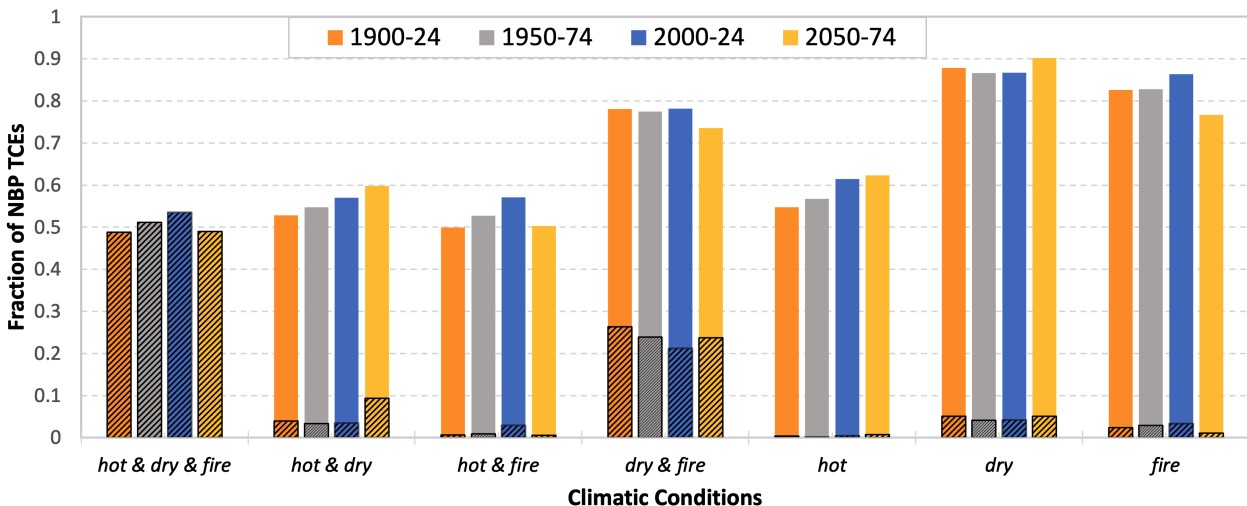

**Figure 5.** Fractional distribution of carbon cycle time-continuous extremes (TCEs) driven by compound climate drivers at lag of 1 month. The unhatched and hatched bars represent the mutually inclusive and exclusive compound and individual climate drivers, respectively. The exclusive climate drivers are always less than or equal to mutually inclusive drivers. The different colored bar represents following periods: 1900–24, 1950–74, 2000–24, and 2050–74 (*from left to right bar*). Most carbon cycle extremes are driven by interactive effect of climate drivers.

The effect of rising temperature on vegetation growth and carbon uptake is dependent on the geographical location. Pan et al. (2019) found that net ecosystem production had a negative sensitivity to warming across 81% of the global vegetated land area during 2007–18 and only the higher latitudes and Tibetan Plateau (TIB) had a positive sensitivity of NBP to temper-
ature. Similarly, Marcolla et al. (2020) found a positive sensitivity of NBP to air temperature in higher latitudes and negative sensitivity in the tropics. Since the tropics have the largest standing biomass and high latitude regions have the largest stored carbon, understanding the contrasting impacts of climate change across these regions is important to understanding climate–carbon feedbacks. The next two sections will briefly discuss the changing characteristics of extremes in the tropics and at high latitudes.

**3.4   Increasing temperature sensitivity and weakening terrestrial carbon sink across the tropics**

Observation-based studies have reported a decline in the rate of carbon uptake in Amazonian forests, and similar declines in the African tropics are expected in the future (Hubau et al., 2020). Over long timescales, the rising atmospheric $CO_2$ concentration may not necessarily lead to an increase in plant biomass (Walker et al., 2019) as respiration losses outpace carbon uptake. Increasingly frequent and stronger heatwaves, droughts, and fires due to climate change are likely to cause the growth rate of
NBP to flatten by the late 21$^{st}$ century (Figure S9). They may lead to an eventual reduction in total stored carbon and a potential reversal for tropical vegetation from a net carbon sink to a carbon source. Toward the end of the 21$^{st}$ century (2075–99), most





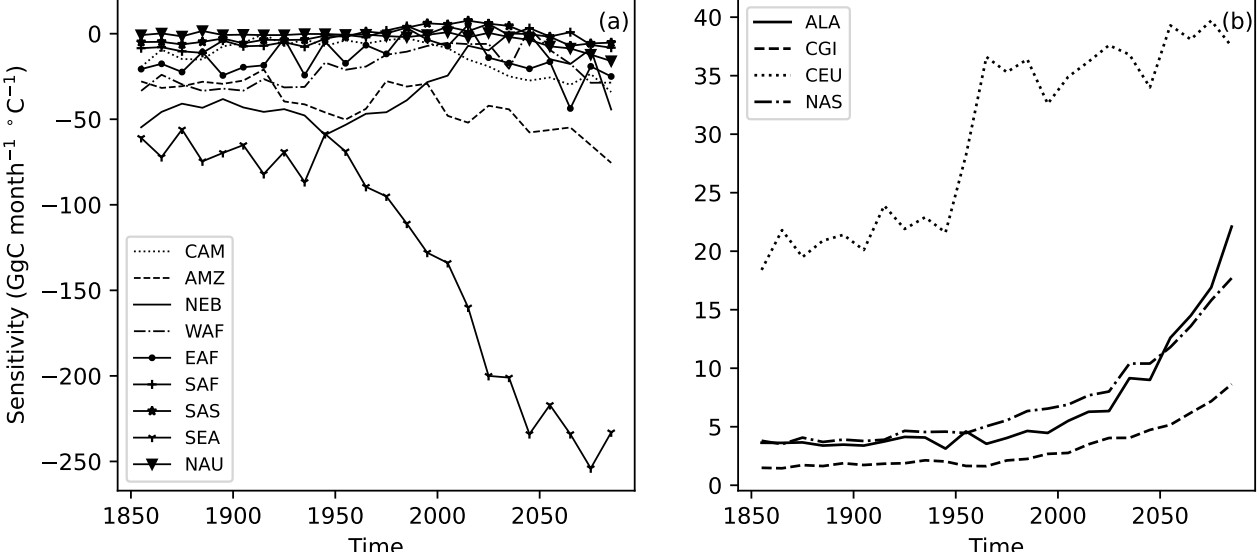

**Figure 6.** Changing temperature sensitivity of detrended anomalies in NBP to detrended anomalies in surface temperature for 10 year time periods at multiple SREX regions. The regions at low latitudes (a) have negative NBP sensitivity to temperature anomalies and high latitudes, and (b) have positive sensitivities.

of the SREX regions (23 of 26) were dominated by negative NBP extremes (Figure 3(b)), especially in the tropical regions (CAM, AMZ, NEB, WAF, EAF, SAF, SAS, SEA, and NAU) (Figure 2). During 2075–99, almost all tropical SREX regions (with the exception of NAU) were dominated by negative NBP extremes.

Rising temperatures and atmospheric $CO_2$ lead to an increasing trend for gross primary production (GPP) and net primary production (NPP) across most of the tropics. However, they are often also accompanied by increases in disturbance (such as droughts and fire), inducing plant mortality and increases in heterotrophic respiration that contribute to significant quantities of negative carbon fluxes from the ecosystem. With the exception of AMZ and SEA that continued to witness an increase in NBP in the model simulation, most of the tropical regions showed a saturation or decline in NBP toward the end of 21st

century (Figure S9). Analysis of temperature trends across tropical regions showed a significant trend toward warmer temperatures during warm (increase in 90th quantile) as well as cool (increase in 10th quantile) months of the year (Figure S7). Rising daily temperatures hinder net carbon uptake by enhancing stomatal closure and ecosystem respiration (Figure S9). The strength of 10 years of negative temperature sensitivity of NBP (see Section S01) increased over time (Figure 6(a)), suggesting an accelerated reduction in NBP growth with rising temperatures. The negative sensitivity values gradually increased

from $-20$ GgC/month·°C to $-33$ GgC/month·°C for CAM, and $-30$ GgC/month·°C to $-70$ GgC/month·°C for AMZ during 1850–2100. South-East Asia (SEA) saw the highest negative temperature sensitivity of $-207$ GgC/month·°C by the end of 21st century. Similar patterns were seen in other tropical regions, suggesting an increasing negative temperature sensitivity of terrestrial ecosystem productivity to carbon uptake in a warming world (Figure 6(a)).





### 3.5 High latitude ecosystems can potentially become sources of carbon under warming climate

High latitude ecosystems store large amounts of carbon below ground, and increasing exposure to warming and disturbance pose the risk of release of stored soil carbon (Marcolla et al., 2020) into the atmosphere. Warmer temperatures at high latitudes create favorable conditions for longer growing seasons, enhanced plant growth and overall increases in greening. All high latitude regions (ALA, CGI, CEU, NAS) showed a trend of positive and increasing NBP sensitivity to changes in air temperature (Section S01) over 1850–2100 (Figure 6).

While the overall impact of warming at high latitudes is expected to increase plant productivity and carbon uptake, high temperature anomalies increasingly induce negative NBP TCEs toward the end of the 21st century. The negative responses of NBP to warm air temperature anomalies were found to occur most frequently during the summer months of July and August. For example, the 90th quantile temperature increased from 13°C by 8°C to 21°C in the NAS region, while similar increases were observed for ALA, CGI and CEU (Figure S8(d)). With warming temperature trends, these periods of carbon losses in
response to temperature extremes have an oversized impact on the overall carbon budget of high latitude ecosystems. Toward the end of the 21st century, CGI and NAS showed strong declines in NBP, becoming a net source of carbon.

The accelerated warming of winter temperatures have large consequences for respiration losses in the Arctic and Boreal regions (Natali et al., 2019; Jones et al., 1998; Commane et al., 2017). Natali et al. (2019) found that the total carbon loss from wintertime respiration in the Arctic was 60% larger than the summer carbon uptake during 2003–17, driven primarily by higher
soil and air temperatures. Contrary to in situ observations, which show significant $CO_2$ emissions at subzero temperatures, the current generation of process-based models shut off the respiration at subzero temperatures, thus underestimating the carbon losses during winter (Natali et al., 2019). The simulation we analyzed showed a 1.7 times higher increase in winter air temperature (10th quantile) compared to summer air temperature (90th quantile) at high latitudes (Figure S8). For example, in NAS, the 10th quantile temperature increased from −25°C during 1900–24 by 14°C to −11°C during 2075–99. The 90th quantile
temperature increased from 13°C by 8°C to 21°C quantile over the same time period (Figure S8(d)). This enhanced rate of warming, especially during winter, resulted in rising wintertime total ecosystem (autotrophic and heterotrophic) respiration, turning some regions to net sources of carbon (Figure S10).

Increases in warm and cold season temperatures induce a potential risk of losing a carbon sink and an accelerated release of stored carbon into the atmosphere. The increase in heterotrophic respiration is likely due to increased thaw of permafrost
(Turetsky et al., 2020), a larger litter pool due to accelerated NPP, and higher microbial decomposition during the extended warm season. As a result, the peak of NBP and NEP started to sharply decline toward the end of 21st century. The CGI region is expected to become a carbon source by the year 2100, and the NBP of the CEU region was gradually decreasing after 1975. The NAS region has shown a reduction in total NBP during 2075–2099, breaking the consistently increasing trend since 1850. With accelerated rising winter temperatures (Figure S8), declining NEP and NBP (Figure S10) and underestimation of
respiration in the current process models, the losses in carbon uptake in the Arctic and at high latitudes in general are expected to be higher in the future.





## 4    Conclusions

The increasing frequency of climate change-driven extremes—such as fire, drought, and heatwaves—have the potential to cause large losses of carbon from terrestrial biomass and soils. The increasing frequency and magnitude of negative NBP

extremes and saturation of NBP toward the end of the 21$^{st}$ century suggests that terrestrial ecosystems may increasingly lose the ability to sequester anthropogenic carbon and ameliorate the impact of climate extremes and change. Under a changing climate, parts of the globe are expected to experience enhanced vegetation growth and positive extremes in NBP; however, they are far outpaced by the frequency and intensity of negative extremes and associated losses in NBP. At the global scale, reductions in deforestation and enhanced $CO_2$ fertilization lead to an increase in NBP. The globally integrated NBP in the

CESM2 reached a peak around 2070 and followed by a large decline toward the end of the 21$^{st}$ century. These losses in NBP were particularly large in the carbon-rich tropical region, followed by arid and semi-arid regions of the world. During 2075–99, 23 out of 26 SREX regions were dominated by negative NBP extreme events, especially in tropical regions. The increasing intensity and magnitude of negative extremes in NBP toward the end of the 21$^{st}$ century and beyond could lead to widespread declines in vegetation, loss of terrestrial carbon storage, and increasingly turn terrestrial ecosystems into a net source of carbon.

Extremes in the carbon cycle, driven by the extremes in environmental conditions, impact vegetation health and productivity. We analyzed anomalies in three primary environmental drivers (*hot, dry and fire*) of NBP extremes. Negative anomalies in soil moisture, causing widespread droughts and water stress in vegetation, were identified as the most dominant driver of negative NBP extremes, affecting almost half of the grid cells experiencing NBP extremes. The interactive and compounded impact of simultaneous anomalies in multiple drivers have especially large impacts on vegetation productivity, beyond the individual

impacts of the variables. Extreme temperature anomalies compounded with dry conditions impact vegetation productivity, more than the sum of individual temperature and moisture anomalies. They also increase the risk and occurrence of fires. The compound effect of all three climate drivers (*hot, dry and fire*) cause the largest fraction of NBP TCEs.

In the tropics, the growth rate of NBP was decreasing, while the magnitude of negative extremes in NBP and the negative temperature sensitivity of NBP was strengthening over time. Large standing carbon stocks (fuel load) with hot and dry climate

(fire weather conditions) increases the fire risk and potential loss of carbon stock during negative NBP extremes.

In the northern high latitudes, accelerated warming leads to permafrost thaw and release of belowground carbon, increasing the likelihood of reversal of the ecosystem to a net source of carbon over time.

This study analyzed climate-driven NBP extremes using an Earth system model simulation from 1850 to 2100. Future work should use multi-model analysis to evaluate the agreement among different Earth system models about the magnitude,

frequency, and spatial distribution of NBP extremes and their attribution to individual and compound climate drivers. Longer-term simulations are needed to analyze the climate–carbon feedback post-2100, when the difference between the rate of $CO_2$ emissions and terrestrial carbon uptake is expected to increase.



*Code availability.* Data analysis was performed in Python, and the analysis codes will be publicly available on GitHub at https://github.com/sharma-bharat/Codes_NBP_Extremes upon acceptance of manuscript.

*Data availability.* The data used here are from the CMIP6 simulations performed by the various modelling groups and available from the CMIP6 archive maintained by from Earth System Grid Federation (ESGF) https://esgf-node.llnl.gov/search/cmip6.

At the CMIP6 archive site https://esgf-node.llnl.gov/search/cmip6 searching for a given model, a given experiment, and a given variable name will yield the link to the dataset that can be downloaded.

*Author contributions.* BS designed the study with inputs from JK, FH, and AG. BS developed the statistical analysis methodology and codes, 390 performed the data analysis, and wrote the manuscript with input from all co-authors.

*Competing interests.* The authors declare that they have no conflict of interest.

*Acknowledgements.* This research was supported by the Reducing Uncertainties in Biogeochemical Interactions through Synthesis and Computation (RUBISCO) Science Focus Area, which is sponsored by the Regional and Global Model Analysis (RGMA) activity of the Earth & Environmental Systems Modeling (EESM) Program in the Earth and Environmental Systems Sciences Division (EESSD) of the 395 Office of Biological and Environmental Research (BER) in the US Department of Energy Office of Science. This research used resources of the National Energy Research Scientific Computing Center (NERSC), a U.S. Department of Energy Office of Science User Facility located at Lawrence Berkeley National Laboratory, operated under Contract No. DE-AC02-05CH11231 for the Project m2467.

We acknowledge the World Climate Research Programme, which, through its Working Group on Coupled Modelling, coordinated and promoted CMIP6. We thank the climate modeling groups for producing and making available their model output, the Earth System Grid 400 Federation (ESGF) for archiving the data and providing access, and the multiple funding agencies who support CMIP6 and ESGF. We thank DOE's RGMA program area, the Data Management program, and NERSC for making this coordinated CMIP6 analysis activity possible.





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
