# Peer review of "Carbon Cycle Extremes Accelerate Weakening of the Land Carbon Sink in the Late 21st Century"

_Biogeosciences, 2022_

## Referee Comment (RC2)

018-0156-3, 2018.

[referee-annotated manuscript omitted]

---

## Author Comment (AC1)

**Legend:**

Text: Reviewer's comments

Text:  My responses

Text: Tracked Changes

Open Access Paper (submitted to EGU):
https://editor.copernicus.org/index.php?_mdl=msover_md&_jrl=11&_lcm=oc108lcm109w&_acm=get_comm_sup_file&_ms=106043&c=233471&salt=1224728252505037330

Supplementary to Paper:
https://bg.copernicus.org/preprints/bg-2022-178/bg-2022-178-supplement.pdf

Revised Paper:   📄 Revised_manuscript.pdf

Tracked Changes:   📄 Tracked_Changes.pdf

In this manuscript, the authors assessed the impacts of climatic extremes on terrestrial carbon budget, based on the simulation with the CESM2. They analyzed top-5% extreme net biome production (NBP) through historical to future periods, and showed an increasing trend of negative extreme NBP, especially in tropical regions. They attributed the trend to factors and found that compound hot, dry, and fire events had a strong impact.

General comments

This study is well focused on detection and attribution of the impacts of extreme conditions on terrestrial carbon budget, such that the trends of extreme-related carbon budget shown in Figure 1 look convincing. The prevailing impact of soil moisture is reasonable, although it could not be independent from precipitation variability. Several results of regional anomalies and dominant factor were remarkable for me. For example, large loss of NBP in East Asia, as much as 3 Pg C (Figure 3), is surprising, because this region is outside of the tropics. Also, I was impressed by the strong impacts of fire on NBP anomalies in many tropical to temperate regions (Fig. 4).

As stated by the authors, this study used only one model (CESM2), and then uncertainty associated with multiple models were not included. I guess that the authors could analyze output data of other models in the CMIP6, but I agree that this remained for forthcoming studies. Similarly, I felt a bit uncomfortable about the use of sole SSP585 result, because this scenario itself is a kind of extreme case. Nevertheless, I found merits in this study and recommend major revisions.

Response: We investigated climate-carbon feedbacks in detail for the SSP585 pathway because the literature suggests (Schwalm et al. 2020, Abadie et al. 2020, Trugman et al. 2018, Park et al. 2015) that this pathway is possibly the best match till 2050 under the current and stated policies and with (likely) plausible levels of $CO_2$ emissions in 2100. In future studies, we will include multiple scenarios to investigate changing climate-carbon variability.

Thank you for finding our paper meritorious and providing us the opportunity to further improve it.

Specific comments

Line 1: Abstract. Please give a short sentence explaining methodology used in this study.

Response: We have revised our abstract and included a short description of the methodology. Please see Tracked Changes: ln 3-15 or the text below:

Using the percentile threshold on the probability distribution curve
of NBP anomalies, we computed negative and positive extremes in NBP.

. . .

Using regression analysis, we found soil moisture anomalies to be the
most dominant individual driver of NBP extremes. The compound effect
of hot, dry, and fire caused extremes at more than 50% of the total
grid cells.

Figure 2: I guess that this figure shows total NBP for each region, i.e., not only extreme NBP but also NBP of usual conditions.

Response: Figure 2 shows the sum of positive and negative carbon cycle extremes during NBP extremes for 25-year time windows for the period 1850-2100. It shows the strength of negative and positive NBP extremes (PgC) across SREX regions over time. The regions in orange color show net carbon cycle extremes and regions in purple color show positive extremes. However, Figure S4 shows the integrated NBP of SREX regions over time. We have revised the caption on Figure 2 (see below) to make it clearer for the readers [Tracked Changes: page 9].

Figure 2. The figure shows the sum of the magnitude of positive and
negative NBP extremes during 25 year periods. The figure shows the
total integrated net impact of carbon cycle extremes (PgC) across
SREX regions for the following periods: (a) 1850-74, (b) 1900-24, (c)
1950-74, (d) 2000-24, (e) 2050-74, and (f) 2075-99. A net gain in
carbon uptake during extremes is represented by a purple color and a
+ sign, and a net decrease is represented by an orange color and a -
sign. For most regions, the magnitude of negative NBP extremes or
losses in carbon uptake were higher than positive NBP extremes or
gains in carbon uptake.

Line 233: I agree with the mechanism but am unsure whether CESM2 has corresponding root structure.

Response: According to the Technical Documentation of CLM5.0 (Lawrence et al. 2018), the model simulates water exchange across the root structure that varies with the soil depth and plant functional type. The soil water flux is dependent on hydraulic conductivity and hydraulic potential among various soil layers via Darcy's Law. Due to the differences in hydraulic properties of soil layers, their soil water content varies by soil depth. The root-soil conductivity depends on evaporative demand and varies by soil layer and is calculated based on soil potential and soil properties, via the Brooks-Corey theory. The rooting depth parameterizations were improved in CLM5.0 with a deepened rooting profile for broadleaf evergreen and broadleaf deciduous tropical trees (Lawrence et al. 2019).

Line 248: Again, I am unsure how CESM2 simulated the post-fire recovery.

Response: There is no explicit post-fire recovery in CLM5.0. It loses biomass from the impact of fire that is possibly restored in subsequent years. CLM5.0 simulates recovery from fire based on the post fire carbon pool and rate of carboxylation among several other factors that govern the plant growth. CLM5.0 does not have post fire vegetation succession.

Line 316: Why Southeast Asia showed such high (much higher than Amazon and Africa) negative sensitivity to temperature? Please explain the underlying reason.

Response: We performed the sensitivity analysis of other carbon cycle fluxes such as GPP (Fig 2.2), Autotrophic respiration (RA: Fig 2.3), and Heterotrophic respiration (RH: Fig 2.4) to temperature. Comparison of these sensitivities for regions of Amazon (AMZ) and Southeast Asia (SEA) show that:

- Rate of increase of negative temperature sensitivity to GPP and RA increased at a high rate in SEA compared to AMZ or other regions from 1850 till 2100. However, AMZ and SEA have similar GPP sensitivity to temperature towards the end of the 21st century.
- The difference of RH sensitivity to temperature between AMZ and SEA was about 150 GgC/month.°C which is large and results in larger negative NBP sensitivity to temperature for SEA than AMZ.

Our results are consistent with the findings of Pan et al. (2020), which investigated the NPP and RH sensitivity to temperature across SREX regions and found that SEA had the highest NPP and RH sensitivity to the temperature.

Your comment highlights an important point and we have revised our manuscript to add additional discussions [Tracked Changes: lines 333-338 or text below].

The negative sensitivity values gradually increased from -20 GgC/month.°C to -33 GgC/month.°C for CAM, and -30 GgC/month.°C to -70GgC/month.°C for AMZ during 1850-2100. South-East Asia (SEA) saw the highest negative NBP sensitivity of -207 GgC/month.°C to temperature by the end of the 21st century. The possible reasons for the large difference in the NBP sensitivity for the region of SEA compared to other tropical regions, e.g. AMZ, are the higher rate of decline in GPP sensitivity to temperature and the highest heterotrophic respiration (RH) sensitivity to temperature of about 90 GgC/month.°C for the region of SEA. Our findings were consistent with Pan et al. (2020), who analyzed seven Terrestrial Biosphere Models and found that the region of SEA had the largest negative NPP sensitivity and positive RH sensitivity to temperature.

[Figure]

Fig 1.1: Temperature Sensitivity to NBP

[Figure]

Fig 1.2: Temperature Sensitivity to GPP

[Figure]

Fig 1.3: Temperature Sensitivity to RA

[Figure]

Fig 1.4: Temperature Sensitivity to RH

References:

Lawrence, D. M., Fisher, R. A., Koven, C. D., Oleson, K. W., Swenson, S. C., Vertenstein, M., et al. (2018). Technical Description of version 5.0 of the Community Land Model (CLM). February 2018. https://www.cesm.ucar.edu/models/cesm2/land/CLM50_Tech_Note.pdf

Lawrence, D. M., Fisher, R. A., Koven, C. D., Oleson, K. W., Swenson, S. C., Bonan, G., et al. (2019). The Community Land Model version 5: Description of new features, benchmarking, and impact of forcing uncertainty. Journal of Advances in Modeling Earth Systems, 11, 4245– 4287. https://doi.org/10.1029/2018MS001583

Pan, S., Yang, J., Tian, H., Shi, H., Chang, J., Ciais, P., et al. (2020). Climate extreme versus carbon extreme: Responses of terrestrial carbon fluxes to temperature and precipitation. Journal of Geophysical Research: Biogeosciences, 125, e2019JG005252 https://doi.org/10.1029/2019JG005252

Schwalm, C.R.; Glendon, S.; Duffy, P.B. RCP8.5 tracks cumulative CO2 emissions. Proc. Natl. Acad. Sci. USA 2020, 117, 19656–19657. https://doi.org/10.1073/pnas.2007117117.

Luis M. Abadie, Luke P. Jackson, Elisa Sainz de Murieta, Svetlana Jevrejeva, Ibon Galarraga, Comparing urban coastal flood risk in 136 cities under two alternative sea-level projections: RCP 8.5 and an expert opinion-based high-end scenario, Ocean & Coastal Management, Volume 193, 2020, 105249, ISSN 0964-5691, https://doi.org/10.1016/j.ocecoaman.2020.105249.

Trugman, A. T., Medvigy, D., Mankin, J. S., & Anderegg, W. R. L. (2018). Soil moisture stress as a major driver of carbon cycle uncertainty. Geophysical Research Letters, 45, 6495– 6503. https://doi.org/10.1029/2018GL078131

Chang-Kyun Park, Hi-Ryong Byun, Ravinesh Deo, Bo-Ra Lee, Drought prediction till 2100 under RCP 8.5 climate change scenarios for Korea, Journal of Hydrology,  Volume 526, 2015, Pages 221-230, ISSN 0022-1694, https://doi.org/10.1016/j.jhydrol.2014.10.043.

---

## Author Comment (AC2)

**Legend:**

Text: Reviewer's comments

Text:  My responses

Text: Tracked Changes

Open Access Paper (submitted to EGU):
https://editor.copernicus.org/index.php?_mdl=msover_md&_jrl=11&_lcm=oc108lcm109w&_acm=get_comm_sup_file&_ms=106043&c=233471&salt=1224728252505037330

Supplementary to Paper:
https://bg.copernicus.org/preprints/bg-2022-178/bg-2022-178-supplement.pdf

Revised Paper:  📄 Revised_manuscript.pdf

Tracked Changes:  📄 Tracked_Changes.pdf

The paper investigates extremes in NBP based on CESM2 simulations. It uses a rather rough spatial resolution - the 26 SREX regions - and monthly timesteps. No comparison to observations are used (taken for granted that CESM2 historical runs represent "reality" faithfully) and considers only a single scenario - SSP5-8.5.  These two are severe limitations for the paper; in particular, extreme events are likely to be very sensitive on both model and scenario choice. Only at the end is an opening for multi-model and scenario runs, but they are strongly recommended already for this paper.

Response: We computed the anomalies of NBP at every grid cell using the monthly time series data (see lines 100-101). The extremes were computed based on the global NBP anomalies (see lines 88-90). The attribution of NBP extremes to climate drivers was also performed at every grid cell (see lines 136-137). Results were aggregated and presented for 26 SREX regions for regional analysis to compare across regions.

Discussions on the merits of using a single model and multimodel ensemble:

Predictive understanding of climate, specifically in the context of carbon cycle, can benefit potentially in different ways through analysis of multiple Earth system model (ESM) simulations vs. a deep dive into the simulations from a single model. Multi-model statistics tend to provide a better predictive understanding of trends and patterns in climate statistics including variability and predictability. However, aggregate multi-model statistics tend to lose physical consistency and often preclude our ability to investigate model processes and parameterizations and suggest ways to improve them. In this paper, we chose to use one of the established ESMs, specifically CESM2, to better understand the predictive ability and strength and weakness of the climate-carbon feedbacks of this community ESM. Furthermore, out of the current generation (Coupled Model Intercomparison Project phase 6: CMIP6) of ESMs, only two models, CESM2 (which we used here) and CNRM-ESM2-1, produced all the simulation outputs (including "fFireAll": "Carbon Mass Flux into Atmosphere Due to $CO_2$ Emission from Fire Including All Sources") that can be used for attribution studies in this paper, but even out of those two, only CESM2 currently produces a comprehensive set of simulations outputs and better represents a wider range of feedback processes when compared with observations. We wanted to investigate the science of one model and investigate in detail the anomalies, climate-carbon feedbacks across space and time, understand the mechanisms embedded in this model, and what are the possible implications of our findings. Investigating one model in detail also helped us identify some of the model artifacts. For example, we saw that the magnitude of extremes in NBP and interannual variability in NBP increased drastically during the period 2000-15. We reached out to the modeling group and found that during this period the LULCC forcing was changed from decadal to annual, which likely increased the carbon-climate variability during 2000-15.

For the variables that we analyzed in this study, CESM2 is always ranked among the top 3 ESM based on the ILAMB benchmarking scores for the historical period, see Figure 5.22 of Chapter 5: Global Carbon and other Biogeochemical Cycle and Feedbacks, IPCC Sixth Assessment Report Working Group 1 (Canadell et al. 2021).

We investigated climate-carbon feedback in detail for the SSP585 pathway because the literature suggests (Schwalm et al. 2020, Abadie et al. 2020, Trugman et al. 2018,  Park et al. 2015) that this pathway is possibly the best match to midcentury under the current and stated policies and with (likely) plausible levels of $CO_2$ emissions in 2100.

Based on the reviewers' comments we have expanded the caveats section to include these limitations (see below or Tracked Changes: ln 400-403). In the future, we will test the multi-model variability and their physical consistency. In the next study, we plan to use multi-model and multi-scenario analysis and address the issues raised by the reviewer.

```
This study analyzed climate-driven NBP extremes using one Earth
system model, CESM2, from 1850 to 2100. Using only CESM2 simulations
helped us to delve deeper into the climate-carbon feedbacks across
different periods and spatial resolutions, as well as to identify
model artifacts. However, the current study lacks comparison to
observations, other Shared Socioeconomic Pathways, and other Earth
System Models.
```

The authors do a good job in preparing the input, i.e. the calculation of anomalies, by taking out the annual cycle using SSA. The reviewer isn't that happy with the decision to define every SSA component with a dominant period > 10 years as "nonlinear trend". This is arbitrary and contrary to our knowledge of long-term cycles in observations, e.g. of Sea Surface Temperatures.

Response: Thank you for acknowledging the data preparation methodology. The trends in this study were defined as the sum of all signals from SSA with a return period of 10 years and higher at every gridcell. Therefore, detrending removes all periods that are greater or equal to 120 months. We chose periods larger than 10 years for defining trends because we wanted ENSO, which has a return period of 3 to 7 years, to be part of anomalies. The trends are non-linear for most grid cells because the relationships between photosynthesis and elevated $CO_2$ and temperature are not linear. Sharma et. al. 2022 shows the trends of GPP (Figure 2). Although we did not perform statistical tests to check for nonlinearity, we can qualitatively say that the trends in the carbon cycle are nonlinear. Zscheischler et al. (2013, 2014) used SSA to calculate anomalies and defined non-linear trends as the sum of all signals from SSA with a return period of 30 years and higher. We defined trends as larger than 10 years consistent with existing literature.

Another issue is while it is true that responses to climate drivers may vary over short time scales ("daily to monthly", l. 55) and CMIP6 simulations are available at daily scales, it is suprising that the authors nevertheless use monthly data only, depriving them from any conclusions on these shorter scales.

Response: While the reviewer is correct that some of the atmospheric variables like temperature and precipitation are available at daily temporal resolution. The land component variables like GPP, NBP, and Fire (fFireAll), that we used in our study, are available only at monthly time resolution (source: https://esgf-node.llnl.gov/projects/cmip6/).

The explanation of compound events and in particular the concept of mutually inclusive / exclusive is confusing. It should be rephrased and in particular simplified.

Response: We have revised the description (see below) of the exclusive and inclusive climate drivers in the revised manuscript to better explain their meaning [Tracked Changes: lines 289-293].

```
Mutually inclusive climate drivers represent the simultaneous
occurrence of various climatic conditions that drive extreme events
in NBP. Mutually exclusive climate drivers are those climatic
conditions that do not occur at the same time to cause an extreme
event. For example, if an extreme event in NBP is driven by both hot
and dry conditions, the mutually exclusive climate driver is only hot
& dry and the mutually inclusive drivers are hot, dry, and hot & dry.
```

Some specific comments:

l. 194: change LULUCC forcing from decadal to annual and back to decadal: do you mean in the model? If so, the net carbon uptake change would just be an artefact of the model setup, which would be embarrassing since no proper conclusions (also for the other 25 year periods) could be drawn.

Response: Yes, it is an artifact of the model setup. We saw an increased magnitude of NBP extremes and interannual during the time period 2000-24. It is most likely due to changes in the LULCC forcing from decadal to annual for the period 2000-15. This increased frequency of LULCC forcing likely led to high variability in NBP. However, the impact of change in LULCC forcing was not as significant for mean NBP changes (Figure S3). Hence, the findings that we have presented are valuable for evaluating the changes in the extremes in NBP over time and

across regions. We have revised the paper to clarify this doubt [Tracked Changes: lines 210-213].

> The large magnitude of net carbon uptake changes during the period 2000-24 was likely due to the change in LULCC forcing from decadal to annual during 2000-2015 and then back to decadal from 2015 onward. The increased temporal resolution of LULCC forcing possibly caused higher climate variability due to biogeophysical feedbacks and subsequently led to increased carbon cycle variability and extremes. Since we focused on NBP extremes, which are tails of PDF of anomalies (or interannual variability) of NBP, the magnitude of carbon cycle extremes was large during this period. However, the impact of LULCC forcing was not as significant on mean NBP changes (Figure S3).

l. 207: "global anomalies": are you sure - the NBP TCEs are surely based on each SREX regions separately? It wouldn't make sense to put thresholds for anomalies worldwide, since some regions would have anomalies all the time, and others never.

Response: Yes, the thresholds were calculated using the global anomalies in NBP. We wanted to quantify the NBP extremes that are significant globally. Moreover, we wanted to perform a comparative analysis of the global NBP extremes across various SREX regions. However, you are correct that in current study some regions will show more extremes than others because of larger NBP and interannual variability. We have revised the manuscript to clarify this ([Tracked Changes: lines 96-97; 98-100] and see below).

> We wanted to quantify the NBP extremes that are significant globally and compare the distribution of global NBP extremes across various regions. The Intergovernmental Panel on Climate Change (IPCC) (Seneviratne et al., 2012) defines extremes of a variable as the subset of values in the tails of the probability distribution function (PDF) of anomalies. Based on the global PDF of NBP anomalies, we selected a threshold value of q, such that total positive and negative extremes constitute 5% of all NBP anomalies.

l. 249: "Hot temperatures that persist for long periods induce heatwaves" - isn't that the very same? Remove the tautology in that case.

Response: Thank you for pointing it out. We have changed it in the revised manuscript [Tracked Changes: line 265] and shown below.

Hot temperatures  over long periods tend to reduce ecosystem production and enhanced terrestrial respiration . . .

l. 264: "Reduction of fuel load by changing vegetation composition...": who does change the composition? In the model? The SSP5-8.5 is the "business as usual" scenarion where no (major) changes (like e.g. forest restructuring) is foreseen. Also, not every change in vegetation composition reduces fuel load. What do you imply here?

Response: We made general qualitative observations that could explain the reason for the decline in the number of SREX regions dominated by fire. However, we have not systematically analyzed LULCC in this current study. Therefore, we deleted this sentence from the revised paper. [Tracked Changes: lines 280-281]

l. 312: enhancing stomatal closure and ecosystem respiration": this is a contradiction. It is possible that plants' response to increased CO2 offer is a partial closing of the stomata, leading to sink saturation, but at the same time, this REDUCES respiration, i.e. the opposite.

Response: Yes, the increased atmospheric concentration of $CO_2$ leads to stomatal closure and reduction in autotrophic respiration. While droughts decrease both vegetation productivity and terrestrial respiration, hot temperatures decrease vegetation productivity and increase terrestrial respiration due to a large increase in heterotrophic respirations (Pan et al. 2020). In models, stomatal closure and ecosystem respiration (sum of autotrophic and heterotrophic respiration) are not linked directly. In Figures S9 and S10 we show the rise in the autotrophic and heterotrophic respirations over time in the tropics and high latitudes.

There are more detailed comments and suggestions to changes in the attached pdf, please consider these as well.

Preferably, the paper should be enlarged in scope by including additional models and scenarios, leading to a major revision. If this is not an option, the other changes required are more of "minor" character.

Response: While including other models does add value it will be a larger task and warrant a new study approach. We think the findings of our paper significantly contribute to increasing our understanding of NBP extremes and their climate drivers over time. In our next study, we will include multi-model and multi-observation comparisons of extremes in carbon fluxes.

Additional Comments are also addressed ([link](link)) and the manuscript is revised accordingly.

References:

Schwalm, C.R.; Glendon, S.; Duffy, P.B. RCP8.5 tracks cumulative CO2 emissions. Proc. Natl. Acad. Sci. USA 2020, 117, 19656–19657. https://doi.org/10.1073/pnas.2007117117.

Luis M. Abadie, Luke P. Jackson, Elisa Sainz de Murieta, Svetlana Jevrejeva, Ibon Galarraga, Comparing urban coastal flood risk in 136 cities under two alternative sea-level projections: RCP 8.5 and an expert opinion-based high-end scenario, Ocean & Coastal Management, Volume 193, 2020, 105249, ISSN 0964-5691, https://doi.org/10.1016/j.ocecoaman.2020.105249.

Trugman, A. T., Medvigy, D., Mankin, J. S., & Anderegg, W. R. L. (2018). Soil moisture stress as a major driver of carbon cycle uncertainty. Geophysical Research Letters, 45, 6495– 6503. https://doi.org/10.1029/2018GL078131

Chang-Kyun Park, Hi-Ryong Byun, Ravinesh Deo, Bo-Ra Lee, Drought prediction till 2100 under RCP 8.5 climate change scenarios for Korea, Journal of Hydrology, Volume 526, 2015, Pages 221-230, ISSN 0022-1694, https://doi.org/10.1016/j.jhydrol.2014.10.043.

Canadell, J.G., P.M.S. Monteiro, M.H. Costa, L. Cotrim da Cunha, P.M. Cox, A.V. Eliseev, S. Henson, M. Ishii, S. Jaccard, C. Koven, A. Lohila, P.K. Patra, S. Piao, J. Rogelj, S. Syampungani, S. Zaehle, and K. Zickfeld, 2021: Global Carbon and other Biogeochemical Cycles and Feedbacks. In Climate Change 2021: The Physical Science Basis. Contribution of Working Group I to the Sixth Assessment Report of the Intergovernmental Panel on Climate Change [Masson-Delmotte, V., P. Zhai, A. Pirani, S.L. Connors, C. Péan, S. Berger, N. Caud, Y. Chen, L. Goldfarb, M.I. Gomis, M. Huang, K. Leitzell, E. Lonnoy, J.B.R. Matthews, T.K. Maycock, T. Waterfield, O. Yelekçi, R. Yu, and B. Zhou (eds.)]. Cambridge University Press, Cambridge, United Kingdom and New York, NY, USA, pp. 673–816, doi: 10.1017/9781009157896.007.

Sharma, B., Kumar, J., Collier, N., Ganguly, A. R., & Hoffman, F. M. (2022). Quantifying carbon cycle extremes and attributing their causes under climate and land use and land cover change from 1850 to 2300. Journal of Geophysical Research: Biogeosciences, 127, e2021JG006738. https://doi.org/10.1029/2021JG006738.

Zscheischler, J., Reichstein, M., Buttlar, J. von, Mu, M., Randerson, J. T., and Mahecha, M. D. (2014), Carbon cycle extremes during the 21st century in CMIP5 models: Future evolution and attribution to climatic drivers, Geophys. Res. Lett., 41, 8853– 8861, doi:10.1002/2014GL062409.

Jakob Zscheischler, Miguel D. Mahecha, Stefan Harmeling, Markus Reichstein, Detection and attribution of large spatiotemporal extreme events in Earth observation data, Ecological Informatics, Volume 15, 2013, Pages 66-73, ISSN 1574-9541, https://doi.org/10.1016/j.ecoinf.2013.03.004.

Pan, S., Yang, J., Tian, H., Shi, H., Chang, J., Ciais, P., et al. (2020). Climate extreme versus carbon extreme: Responses of terrestrial carbon fluxes to temperature and precipitation. Journal of Geophysical Research: Biogeosciences, 125, e2019JG005252 https://doi.org/10.1029/2019JG005252

---

## Author Response (AR1)

Note: *The comments of the reviewer are italicized*, the responses are in regular font, and **revisions in the paper are bold.**
Revised manuscript: 🗋 Sharma_2022_Revised_Full.pdf
Tracked changes: 🗋 Sharma_2022_Tracked_Version.pdf

======================================================================

*Please implement all suggested changes as you suggested them in the author response letters and provide a revised manuscript. You also need to include how your analysis differs from Zscheischler et al. 2014 in the methods and discussion section, i.e. integrate your responses to CC1.*

Response:
While the reply to the CC1 (https://doi.org/10.5194/bg-2022-178-AC3 ) elaborates on the differences between this study and the paper (Zscheischler et al. 2014) of the CC1. We have included the major differences in our methodology and results that are contrary to Zscheischler et al. 2014, please see tracked changes lines 124-129 and 241-244 or the text below:

Revised manuscript (lines 123-127) or Tracked changes (lines 123-129):

**Attribution analysis was performed in recent studies of large connected manifolds of spatio-temporal continuous extremes in the carbon cycle (Sharma et al., 2022b; Zscheischler et al., 2014; Flach et al., 2020) by comparing the medians or mean state of climate driver(s) during or preceding a carbon cycle extreme with climate extreme for the same spatio-temporal region or grid cells affected during carbon cycle extremes. This method may not capture the variability at smaller regional to grid cell scales.**

Revised manuscript (lines 237-240) or Tracked changes (lines 242-245):

**This is contrary to the results of Zscheischler et al. (2014), who found a strengthening of positive (net ecosystem production, NEP) extremes over time using CMIP5 ESMs. However, the ratio of negative to positive carbon cycle extremes in our study lie within the multi-model spread of the relative strength of NEP 240 extremes in CMIP5 ESMs (Zscheischler et al., 2014).**
* * *
*AC1 - your response to comment "Line 233: I agree with the mechanism but am unsure whether CESM2 has corresponding root structure." Please include this explanation in the method section to ensure major model functionalities and process representation are described.*

Based on your and reviewer's recommendations, we have included the following changes in our revised version. Revised manuscript (lines 164-171) or Tracked changes (lines 167-176):

**We expect that CESM2 could simulate the impact of variability of climate drivers on ecosystem processes because the Community Land Model version 5 (CLM5 (Lawrence et al., 2018)), the land model component of CESM2, simulates water exchange across the root structure that varies with soil depth and plant functional type. The soil water flux is dependent on hydraulic conductivity and hydraulic potential among various soil layers via Darcy's Law. Due to the differences in hydraulic properties of soil layers, their soil water content varies by soil depth. The root-soil conductivity depends on evaporative demand and varies by soil layer and is calculated based on soil potential and soil properties, via Brooks-Corey theory. The rooting depth parameterizations were improved in CLM5 with a deepened rooting profile for broadleaf evergreen and broadleaf deciduous tropical trees (Lawrence et al., 2019).**
* * *
*AC2 - your response to comment refering to "line 312: enhancing stomatal closure and ecosystem respiration": this is a contradiction. It is possible that plants' response to increased CO2 offer is a partial closing of the stomata, leading to sink saturation, but at the same time, this REDUCES respiration, i.e. the opposite." Please correct the wording to make clear you mean heterotrophic respiration.*

Based on your and reviewer's recommendations, we have included the following changes in our revised version. Revised manuscript (lines 332-334) or Tracked changes (lines 338-339):

**Rising daily temperatures hinder net carbon uptake by enhancing stomatal closure, under conditions of water stress, and increasing heterotrophic respiration (Figure S9).**
* * *
The earlier feedback from the reviewers and community can be found under the Discussions on the manuscript page. Below are the links to the earlier replies:

- Reply to AC1: https://doi.org/10.5194/bg-2022-178-AC1
- Reply to AC1: https://doi.org/10.5194/bg-2022-178-AC2
- Reply to CC1: https://doi.org/10.5194/bg-2022-178-AC3